# Application of LiDAR Data for the Modeling of Solar Radiation in Forest Artificial Gaps—A Case Study

**Leszek Bolibok [1] and Michał Brach [2],*** 

[1]  Department of Silviculture, Institute of Forest Sciences, Warsaw University of Life Sciences, Nowoursynowska 159, 02-776 Warsaw, Poland; leszek_bolibok@sggw.edu.pl

[2]  Department of Geomatics and Land Management, Institute of Forest Sciences, Warsaw University of Life Sciences, Nowoursynowska 159, 02-776 Warsaw, Poland

*  Correspondence: michal_brach@sggw.edu.pl; Tel.: +48-22-5938-213

**Abstract:** Artificial canopy gaps (forest openings) are frequently used as an element of regeneration cutting. The development of regeneration in gaps can be controlled by selecting a relevant size and shape for the gap, which will regulate the radiation microclimate inside it. Based on the size and shape of a gap computer models can assess where solar radiation is decreased or eliminated by the surrounding canopy. The accuracy of such models to a large extent depends on how the modeled shape of a gap matches the actual shape of the gap. The aim of this study was to compare the results of modeling solar radiation availability by applying Solar Radiation Tools (SRT) that use a different digital surface model (DSM) for a description of the shape of a studied gap, with the results of the analysis of 27 hemispherical photographs. The three-dimensional gap shape was approximated with the use of simple geometrical prisms or airborne laser scanning (LiDAR) data. The impact of two variations of exposure (automatic and manual underexposure) and two variations of automatic thresholding on the congruence of SRT and Gap Light Analyzer (GLA) results were studied. Taking into account information on differences in height between trees surrounding the gap enhanced the results of modeling. The best results were obtained when the boundary of the gap base estimated from LiDAR was expanded in all directions by a value close to a mean radius of the crowns of surrounding trees. Modeling of radiation conditions on the gap floor based on LiDAR data by an SRT program is efficient and more time effective than taking hemispherical photographs. The proposed solution can be successfully applied as a trustworthy source of information about light conditions in gaps, which is needed for management decision-making in silviculture.

**Keywords:** solar radiation; artificial canopy gap; gap shape; LiDAR; hemispherical photography; regeneration cutting

## 1. Introduction

Small oval-shaped forest openings (artificial canopy gaps) with an area of less than 1 ha have for a long time [1] been an important element of regeneration cuttings used for the creation of stands with a diverse age and species structure [2–7]. Relative gap size (the ratio of the gap diameter D to the height of surrounding canopy H) and its shape and orientation can impact the availability of solar radiation at the forest floor [8–10]. In addition to that, the location's latitude [11], slope, and aspect [12,13] further modify the amount of solar energy reaching the gap. The gap heat balance has an impact on spring thawing of snow, soil temperature and moisture, the temperature and humidity of the air, and the occurrence of ground frost. By adjusting gap sizes and shapes it is possible to provide conditions appropriate to the preferences of desired tree species by minimizing the risk of occurrence of ground frost [14] or by creating optimal moisture and light conditions [6,7,15–17]. Knowledge of potential

radiation conditions in gaps of diverse size and shape in a given location can make it easier to select an optimal form of regeneration cuttings. For this reason, models are applied in order to determine potential diversity of radiation conditions on planned regeneration cutting gaps [2,18,19].

The most advanced models describing radiation conditions in gaps employ detailed information about trees growing at the edges and in close vicinity to the gap [20,21]. These models take into account information about crowns' positions in 3D space and about their transmissivity depending on the condition and species of the tree. For such models, high accuracy is, however, inseparable from the time- and labor-intensive activity of gathering information about trees surrounding gaps, which significantly limits their rate of use. In parallel to the development of the above-mentioned class of models, other models were developed to allow a description of solar energy supplies in various fragments of the landscape represented by the digital surface model (DSM) [22]. The geometrical foundations of analyses used in these models are analogous to those applied in the analysis of hemispherical photographs. In both cases, the aim is to establish which fragments of the sky are obfuscated by objects found in the surroundings, but in each case information about the location of these objects derives from a different source [23]. Models implemented in GIS software, such as Solar Radiation Tools (SRT) from the ArcGIS program [24], or r.sun light from the GRASS (Geographic Resources Analysis Support System) program [25], describe radiation conditions within the floor of such hollows with an assumption that gap walls are completely unpenetrated by sunrays and that no light from neighboring smaller openings in canopy reaches the floor of the gap. For this reason, hemispherical photography undoubtedly surpasses the mentioned models in the case of modeling light conditions under the canopy. However, in the case of artificial gaps, this advantage is not always large. In contrast to natural gaps, artificial gaps are usually much bigger, which results from management circumstances, e.g., a larger area is needed for mechanization of soil preparation or to increase logging efficiency, etc. [26], or for the sake of limiting competition from the roots of surrounding tree stands [27,28]. The central part of the sky, which provides the majority of solar radiation energy, is uncovered in most locations within the artificial gap. Trees surrounding the gap mostly cover fragments of the sky closer to the horizon. Gaps receive significantly less radiation from low sky regions because they emit a smaller quantity of energy and because radiation travels a relatively longer distance through the canopy layer. Underestimation of this portion of radiation may constitute a relatively small share in the overall balance. In the case of models that employ DSM, there is a potential for correction of their functioning through more faithful representation of the shape of the top boundary of the artificial gap, created by the tops of the trees surrounding the gap.

The development of laser scanning (light detection and ranging—LiDAR) enables a relatively easy acquisition of a DSM [29], which gives the possibility of describing the top surface of the canopy around and at the edge of the artificial gap in the most reliable way. It may allow better accuracy in assessing radiation conditions in comparison with simplified models because open spaces between treetops that allow penetration of more light to the gap floor are taken into account. Although LiDAR technology is still quite expensive, it is expected that the relation of data quality to the price will soon make its application justifiable from the financial point of view [30]. The use of unmanned aerial platforms (Remotely Piloted Aircraft Systems—RPAS) is an alternative method of obtaining high quality DSMs [31,32]. Unmanned platforms make it possible, first of all, to lower the costs of data acquisition and, secondly, to increase the operational potential in small areas [33].

This study is focused on the assessment of the possible LiDAR data application for determining radiation conditions in artificial gaps. It may be alternative method to analysis of hemispherical photographs. The impact of various methods of representation of three-dimensional gap shape by the use of DSM, including application of LiDAR data, on the results of the modeling of radiation conditions by an SRT model was tested in the study.

The outcome of the analysis of hemispherical photographs taken at selected locations in the gap obtained with the Gap Light Analyzer (GLA) served as a reference for the evaluation of the SRT results. The results of the analysis of hemispherical photographs strongly depend on their exposure

and thresholding method and therefore they are not an absolute point of reference. The impact of two popular methods of exposure and thresholding on the results of the GLA model, and hence on the evaluation of the results of the SRT model, was tested.

The main goal of this study was comparison of the indirect estimation of radiation using the hemispherical photographs (GLA) and the estimation of radiation using the model implemented in ArcGIS (solar radiation tool—SRT) that uses variable delineation of the artificially created gap. On this basis, we proposed a simple and reliable procedure enabling estimation of radiation conditions in artificial gaps which can be easily implemented by ready to use model in popular GIS software.

## 2. Methods

### 2.1. Study Area

The field study was carried out within the Forestry Experimental Station Rogów, in a 102-year-old forest stand dominated by the Scots pine *Pinus sylvestris L.* (51°45′12.72″ N, 20°5′31.50″ E), where numerous artificial gaps were cut. One gap, with the surface of 0.23 ha (relative gap size ratio 2.5), located on a flat part of the area, was selected. It was surrounded by 46 trees with a mean crown radius of 3.19 m (standard deviation 1.36 m) and a mean height of 21.35 m. In the gap, twenty-seven sampling points were marked at distances of ca 5 m from one another in three rows running S-N along with the expected highest diversity of intensity of solar radiation within the gap (Figure 1a). The course of the rows was not in straight lines, because it was adjusted to the burrows created during soil preparation for tree planting. The position of sampling points and the position and height of trees creating the boundary of the gap—*sensu* "extended gap" [34]—were determined with the use of classical surveying methods. The obtained measurements were referenced to the existing surveying grid, which enabled reaching ±0.06 m accuracy of reference points [35].

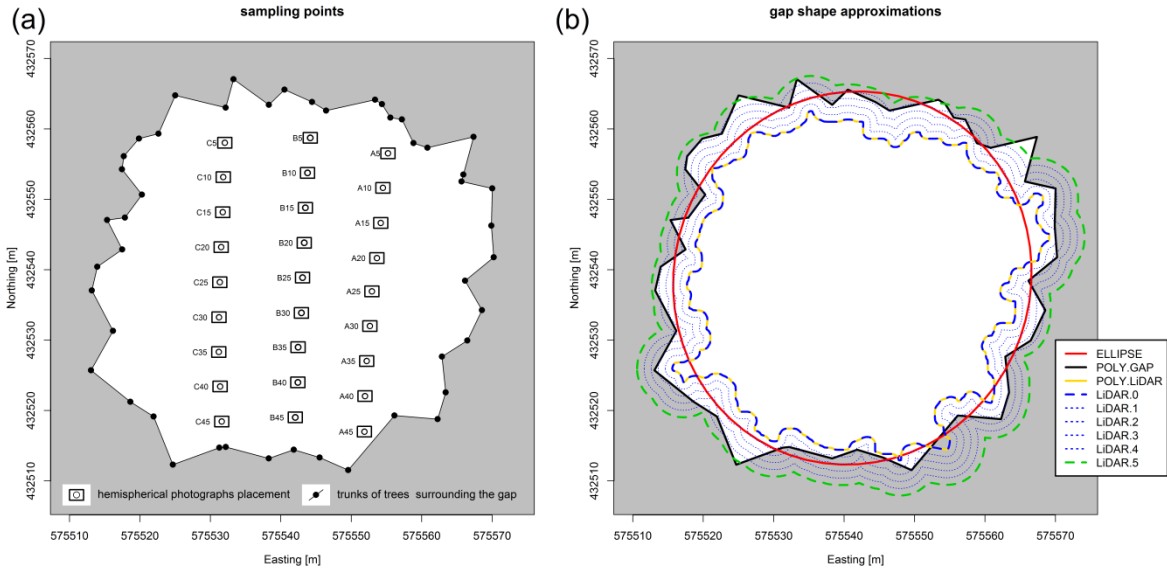

**Figure 1.** Distribution of sampling points within the gap (**a**) and analyzed boundaries of the base of the three-dimensional shape representing the gap (**b**).

### 2.2. Accomplishment of Hemispherical Photographs

A hemispherical photograph was taken at each sampling point with an EOS 5D camera (Canon, Tokyo, Japan) and an 8 mm f/3.5 EX DG circular fisheye lens (SIGMA, Aizu, Japan). Photographs were taken on 28 May 2013 in optimal weather conditions with a completely uniformly overcast sky to avoid problems caused by the presence of the solar disc or clouds standing out from the background. The camera was mounted on a 1.8 m tripod so that the tops of oak saplings covering the floor of the

gap would not be depicted in photographs. The vertical axis was adjusted with a bubble level slotted onto the flash socket of the camera. The camera was oriented with a compass in such a way that the shorter side of the photograph was parallel to the magnetic S-N axis and the top of the photograph was to the north. Aperture priority (F = 8) mode was chosen for taking the photographs and the sensitivity was set to 400 ISO. Photographs were stored as JPG files of 4368 × 2912 pixels.

Two photographs were taken at each sampling point: one with automatic exposure and the second one with underexposure of −2 EV with regard to automatic light measurement Reduction of exposure in comparison to the automatic reading was applied by many authors [36–38] for the sake of reducing errors in the estimation of radiation conditions which can be attributed to improper exposure. In practice, it is less complicated than regulating exposure with reference to readings in an entirely open area [39] or based on a reading from the center of the photograph [40], especially with changing cloud cover and an extended period of time necessary for taking several photographs. Highly diverse recommendations can be found in the literature with regard to underexposure, ranging from −0.5 EV [41] to even −4 EV [42]. This is due to the specificity of the studied forest stands and particular light conditions. After a visual analysis of photographs obtained from earlier trial observations, correction of exposure was set to −2 EV. The whole process of hemispherical data acquisition took 3 h of fieldwork.

## 2.3. LiDAR Data

Laser scanning realized as part of the ISOK (Information System of Country Protection Against Extraordinary Hazards) project was carried out in 2013 in the studied area. The expected accuracy of height positioning for raw point cloud range from 5 to 15 cm [43]. In the case of the analyzed area, the density of point is 13.62 per square meter and the mean distance of points was 0.13 m for vegetation class. A DSM was generated with a resolution of 0.25 m, assuming that the value of each pixel would represent the maximum of the heights of the points it comprises. The DSM based on the maximum value is the best for ensuring correct representation of the crown height model and enabling acquisition of a more uniform surface. Because of a significant difference in elevation between the floor of the gap and the surrounding forest stand canopy (reaching 20 m), those two layers were separated by raster reclassification. Setting relevant class ranges allowed a clear division of the gap edge from the gap interior. Final shape of the boundary was obtained through the application of the boundary clean tool from the ArcGIS package, which applies an algorithm for the smoothing unevenness of different zones of a raster. In essence, its function is to incorporate lower-order pixels into higher-order pixels within a stable window of 9 × 9 pixels. The raster thus obtained was used as a basis for creating boundaries in vector format, which enabled a clear and unequivocal representation of the boundaries of the crowns of the trees surrounding the gap, i.e., of the boundary of the gap. At further stages of this study, the gap boundary was used for creating other variants of gap borders which were obtained by expansion of the original border by a given value. All described procedures are implemented within one model (ArcGIS) very extensively used worldwide and could be relatively simply changed by open source software GRASS.

The LiDAR data from the ISOK project are free for research areas. In the near future, all forest districts in Poland will be covered by laser scanning regularly what is an already a standard procedure for many countries around the world. Taking into account this fact we can assume that this type of data would be easily accessible.

## 2.4. Description of the DSM Used as a 3D Model of a Gap

The three-dimensional (3D) shape of a gap comprising the empty space stretching from the canopy top to the forest floor is very complex. For practical reasons, it is often simplified during modeling of radiation conditions. In this study, a number of methods of modeling gap shape were compared (Figure 2) in order to examine which type of DSM gives the highest congruence with results from the GLA model.

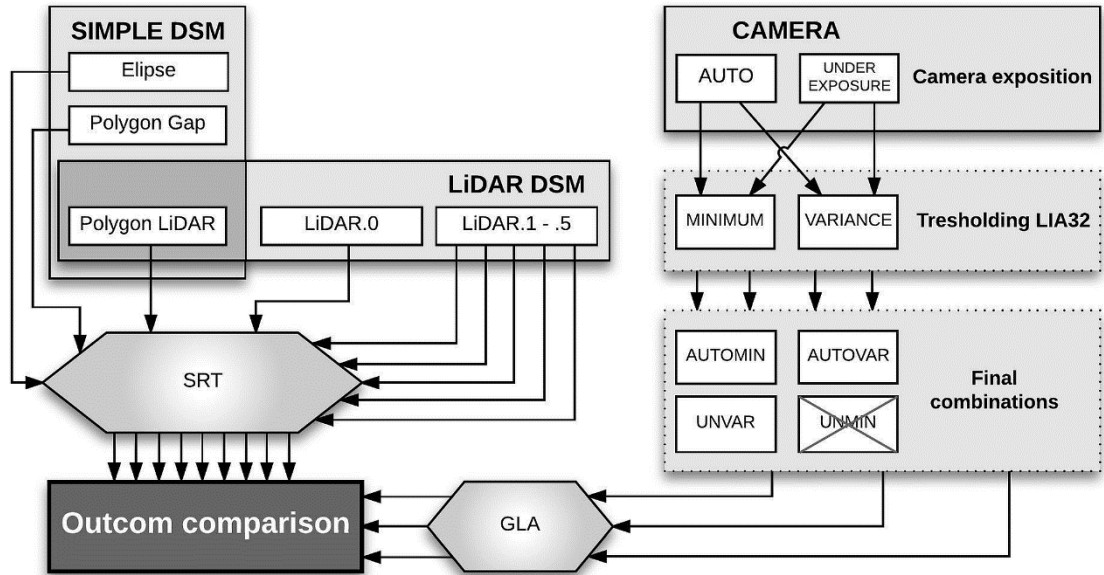

**Figure 2.** Chart representing comparative analysis of radiation conditions in the gap using SRT (Solar Radiation Tool) and GLA (Gap Light Analyzer ) models.

The simplest DSM (ELLIPSE) assumes that a gap is represented by an elliptic cylinder with the height equal to the mean height of trees surrounding the gap (21.35 m) and with an ellipsoid base which is arbitrarily located within the boundaries of the gap marked by the trunks of the surrounding trees (Figure 1b). Although it is a very simplified approach, it was applied in many models [11,18,44–46], and usually artificial clearings made in managed forests often do have the shape close to an ellipse.

The DSM named POLYGON.GAP assumes a 3D gap shape represented as a 21.35 m height block with a polygonal base whose vertices meet the bases of the trees surrounding the gap. This definition of the gap base is consistent with the definition of an extended gap proposed for the description of gaps in natural forests [34]. A comparison of the first and second model allows determining the impact of gap simplification on the modeling results.

The third and fourth DSMs have a polygonal base (Figure 1b) whose boundaries are created according to the interpretation of the extent of the surrounding trees' crowns, determined from the analysis of LiDAR data. The third DSM, (POLYGON.LiDAR), is created according to the interpretation of the extent of the surroundings trees' crowns determined from LiDAR data. Similar to POLYGON.GAP, it assumes that the top boundary of the gap has a constant tree's height equal to the mean height of the surrounding trees. An analysis of this model allows evaluation of the usefulness of gap boundaries marked according to the analysis of the LiDAR point cloud, but without taking into account the height differentiation of the top boundary of the gap. The fourth LiDAR.0 model has the same base as the third one, although the top boundary of the gap is marked by the cross-section of the vertical sides of a block with the surface of the crowns' tops created according to the analysis of LiDAR data. A comparison of results from the POLYGON.LiDAR and LiDAR.0 models allow assessment of how taking into consideration the existence of "hollows" between crowns of particular trees growing at the boundary of the gap can impact the accuracy of radiation conditions modeling.

During preliminary calculations, it turned out that the LiDAR.0 model provides significantly divergent results from the analyses of hemispherical photographs. The opening in the canopy layer above the gap described using LiDAR point cloud analysis seemed "too small", because the results were strongly underestimated. The base of LiDAR.1 was created through geometrical expansion by 1 m in all directions of the polygon that made the base of LiDAR.0, using buffering. The digit used in the name of each subsequent model (LiDAR.1 to LiDAR.5) corresponds to the size of the extension in meters (Figure 1b).

### 2.5. Analysis of Hemispherical Photographs

Results of the analysis of the hemispherical photographs can to a large degree depend on the exposure and method used for photographs thresholding [47–49]. In order to eliminate the subjectivity resulting from manual segmentation of photographs, automatic thresholding with the MINIMUM [50] and VARIANCE [51] algorithms, implemented in the LIA32 program for Win32, were used. Only these two algorithms were used because, in comparison to others implemented in this software, the best managed to distinguish between elements of the canopy top and the sky [48,52]. Information from the blue channel of the photograph stored as a JPG was used for analysis [53]. While processing of hemispherical photographs it was discovered that the MINIMUM algorithm in the case of underexposed photos classifies significant fragments of the sky over the open center of the gap as covered by the tree crowns (black). During the fieldwork when the photographs were taken, the whole sky was overcast; however, differences in the darkness between clouds were big enough for the MINIMUM algorithm to classify the darker ones as fragments of the sky covered by the tree crowns. Analysis of such photographs would produce a decidedly underestimated amount of solar radiation penetrating down to the gap floor. For this reason, only three variations of exposure and thresholding combination will be discussed further: automatic exposure and thresholding using the MINIMUM algorithm (AUTOMIN), automatic exposure and thresholding using the VARIANCE algorithm (AUTOVAR), and photographs underexposed by −2 EV with thresholding using the VARIANCE algorithm (UNVAR) (Figure 2).

After thresholding, photographs were analyzed using the GLA program [54]. A correction with regard to the real projection of the Sigma 8 mm F3.5 EX DG lens was introduced in the program [55]. Also, a correction of local magnetic declination (5°15′ E) was introduced. It was assumed in the model that the clear-sky transmission coefficient in the study area is 0.4 [56]. The SOC (standard overcast conditions) option was selected for modeling of diffuse light supply. The image was analyzed in 45 zenith and 90 azimuth sectors during modeling. The amount of solar radiation energy (both direct and diffuse) reaching the point where the photograph was taken in the vegetation season (between March 31st and November 5th) was calculated by the program with an assumption that the beam fraction was 0.422 and cloudiness index was 0.055.

### 2.6. Modeling of Radiation Conditions in the Gap Using DSM

The Solar Radiation tool implemented in the ArcGIS [23,24] program was used for modeling. The model generated a virtual image of the sky for 88 evenly distributed azimuths (calculation_directions = 88) as it would be seen in terrain described by a given DSM. Insofar as possible, settings of the Solar Radiation Tool (SRT) model were analogous to those in GLA. The Standard Overcast Sky model was selected for modeling diffuse light. The virtual image of the sky (ViewShed) was analyzed using 45 zenith sectors and 90 azimuth sectors during modeling. The transmissivity coefficient for the clear sky was 0.4. The diffuse proportion was 0.6.

Both the GLA and SRT models indicated the same amount of total solar radiation energy above tree crowns (i.e., at the height of 30 m above ground level) in the vegetation season (1003.2 kW h/m$^2$) and identical amounts of direct (423.35 kW h/m$^2$) and diffuse (579.85 kW h/m$^2$) solar radiation energy. Thanks to this parameterization the differences between the two models' outcomes for the height of 1.8 m above ground level (the height at which hemispherical photographs were taken) can be attributed only to differences in how the real boundaries of the gap are described in hemispherical photographs and in the DSM.

### 2.7. Comparison of Modeling Results

During the process of calculating indicators describing the differences between models, it was decided that the results of the GLA model would be treated as a reference point (GLA) to compare the results of the Solar Radiation model (Solar).

Models were compared by calculating the following indicators for 27 sampling points: minimum, maximum and mean absolute error between the models and root mean square error (RMSE) according to Formula (1) in kWh/m$^2$:

$$\text{RMSE} = \sqrt{\frac{\sum_{i=1}^{n}(\text{SRT} - \text{GLA})^2}{n}} \tag{1}$$

where:

GLA = results of the GLA model;

SRT = results of the Solar Radiation model;

N = number of records.

Percent error of the model (PBIAS) which represents the mean tendency for simulated data above or below the values of observed data, expressed in [%], was calculated using Equation (2):

$$\text{PBIAS} = 100 \times \left[ \frac{\sum_{i=1}^{n}(\text{SRT} - \text{GLA})}{\sum_{i=1}^{n}\text{GLA}} \right] \tag{2}$$

The determination coefficient R2 was calculated as the square of the Pearson correlation coefficient of the two models. In addition, the Pearson correlation coefficient was calculated for the relation between the value of differences between the two models for each given sampling point and the distance of the given sampling point to the nearest tree base at the gap boundary.

## 3. Results

### 3.1. Impact of Exposure and Photograph Thresholding Method on Assessment of Congruency of GLA and SRT Models

Canopy openness was chosen as a synthetic indicator of the influence of different hemispherical photography exposure and thresholding scenarios on the assessment of forest canopy properties. It could be defined as the complement of canopy closure (openness = 1 − closure), i.e., the proportion of the sky hemisphere not obscured by vegetation when viewed from a single point [57]. Results of the analysis of the canopy openness assessed at 27 points of the gap with the GLA program are presented in Table 1. Mean values of canopy openness for particular methods of photograph processing differ significantly (Student t test, significantly different means are marked with different letters). Both the extreme and mean values of openness had their highest value in AUTOMIN and lowest in UNVAR variants.

**Table 1.** Impact of a different methods of exposure and thresholding of photographs on the assessment of canopy openness [expressed as a percent of hemisphere area]. Description of methods abbreviations was provided in the text.

| Canopy Openness | AUTOMIN | AUTOVAR | UNVAR |
|---|---|---|---|
| Minimum | 39.48 | 34.42 | 27.94 |
| Mean | 45.19 [a] | 42.15 [b] | 37.43 [c] |
| Maximum | 50.36 | 48.51 | 44.29 |

Means with different letters ([a],[b],[c]) are statistically different.

The obtained results clearly indicate that the implemented method of taking and processing hemispherical photographs has a large impact on the assessment of canopy openness (i.e., the main factor regulating radiation transmissivity at the forest floor) using the GLA model and that it can impact the assessment of congruency of modeling of light conditions in gaps using both GLA and SRT models.

Direct comparison of different variants of GLA and SRT models is presented in Figures 3 and 4 (numerical values of compared parameters are presented in Tables A1 and A2 in the Supporting Information).

For the sake of simplicity the sets of compared modeling results received descriptive names connected in case of results GLA with photographs acquisition and processing mode and in case of SRT results with gap shape model e.g., "UNVAR" means the outcome of GLA model based on photographs taken and processed with UNVAR procedure. All variants of further comparisons and relations between compared models were shown in Figure 2.

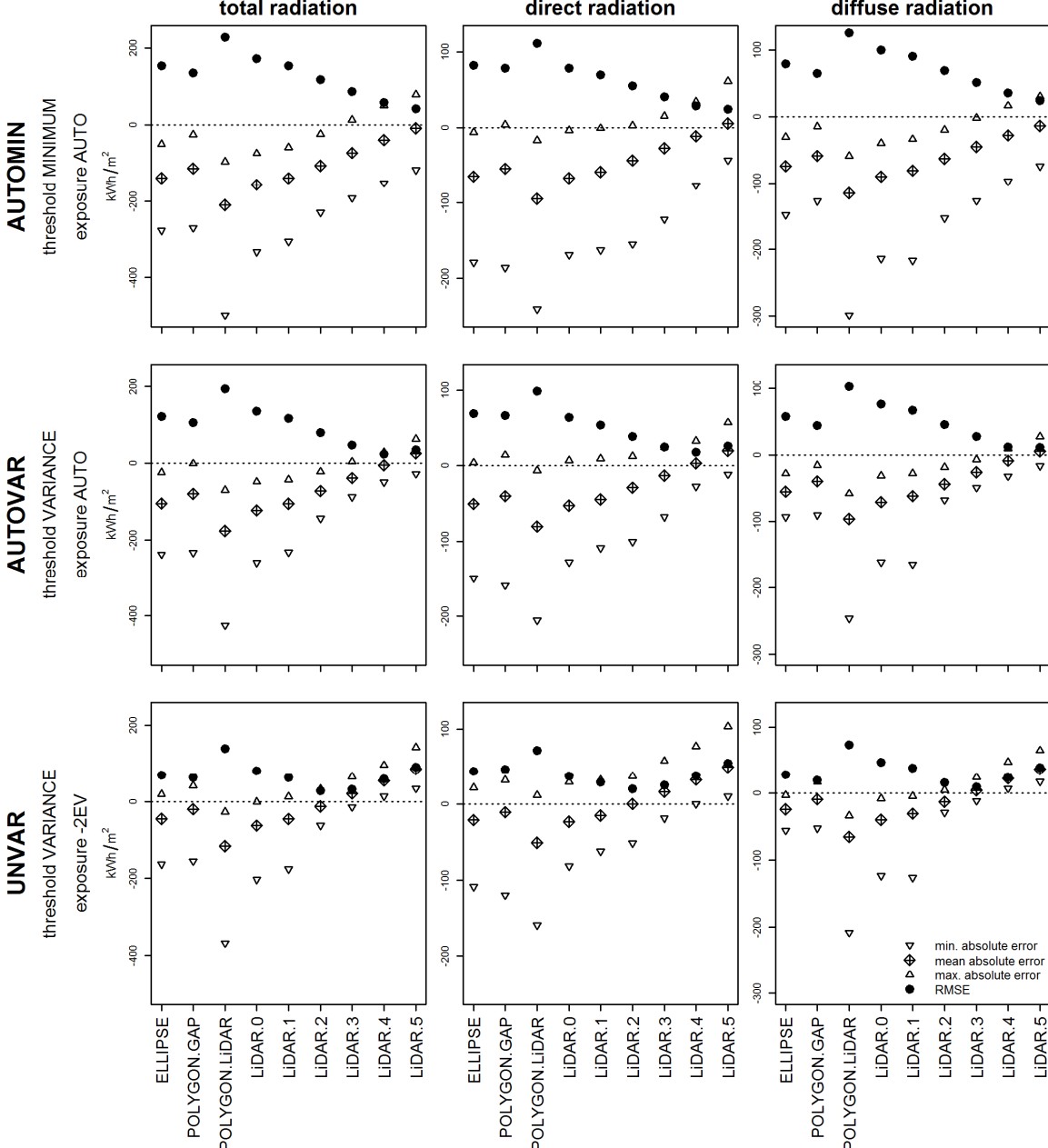

**Figure 3.** Differences between the results of modeling various types of solar radiation (figure columns) within the gap between the SRT model and the results of the analysis of hemispherical photographs using the GLA program in relation to the variants of photograph processing (figure rows) and of gap shape definition.

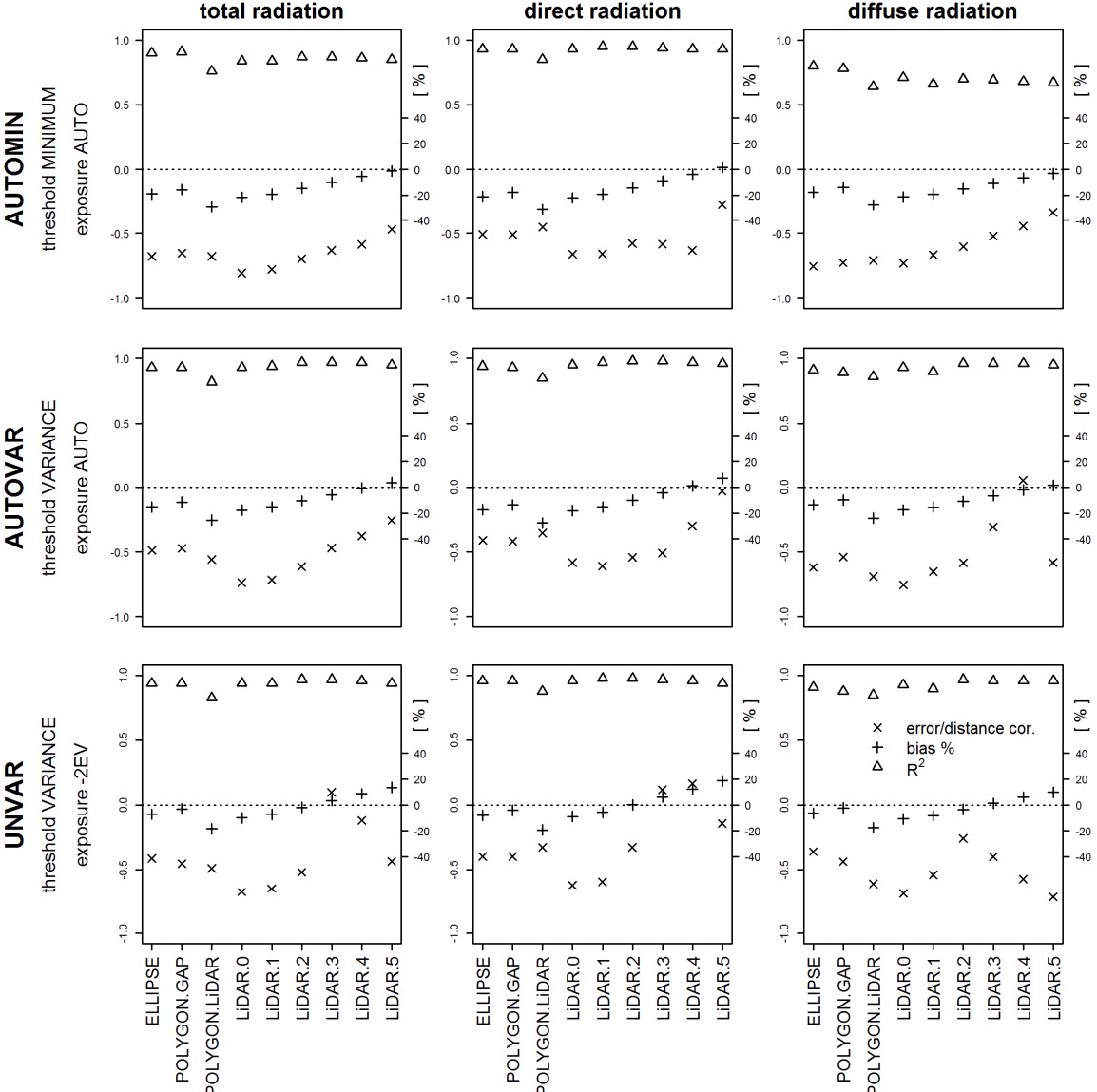

**Figure 4.** Percent error (plus signs) and coefficients of determination (triangles) between the SRT model and results of hemispherical photograph analysis with the GLA program for various types of solar radiation (figure columns) in relation to an applied variant of photograph processing (figure rows) and method of defining gap shape. Correlation between the difference between outcomes of both methods of radiation modeling for sampling points and their distance to the nearest tree at the gap boundary is also marked in the figures (crosses).

The values of the $R^2$ coefficient in AUTOMIN variant are lower (by a few to over a dozen percentage points) than those of AUTOVAR and UNVAR variants, for all gap shapes. The same values differ slightly for AUTOVAR and UNVAR and are between 0.90 and 0.97 for all gap shapes. The highest congruency of the GLA and SRT models was observed in the case of direct radiation modeling and the lowest in diffuse radiation modeling.

The highest divergence of GLA and SRT measured by RMSE and PBIAS values was observed in the AUTOMIN variant. It is, however, difficult to unequivocally indicate which variation of photograph processing provided the highest congruence of GLA and SRT models because it was dependent on the applied method of representation of the gap shape. The advantage of UNVAR was observed for all types of radiation for results obtained in the following gap shape models: ELLIPSE, POLYGON.GAP, POLYGON.LiDAR, LiDAR.0, LiDAR.1, and LiDAR.2. For this group of shape models, PBIAS was

always the smallest for UNVAR and for all methods of photograph processing it had a negative value. In the case of the remaining gap shape models, i.e., LiDAR.3, LiDAR.4, and LiDAR.5, congruence of the GLA and SRT models is the highest for the AUTOVAR variant of photograph processing. The PBIAS coefficients have a negative value for the UNVAR variant for the aforementioned group of gap shape models.

To summarize, regardless of the method of gap shape representation, the AUTOMIN variant of photograph processing resulted in the highest divergence of results of solar radiation modeling in gaps for GLA and SRT models. The UNVAR variant of photograph processing was the best one for the majority of analyzed methods of gap shape representation, although not for all of them.

### 3.2. Impact of Gap Shape Modeling Method on Assessment of GLA and SRT Models

The shape of the line representing RMSE values for each method of gap shape representation is similar for all variants of taking and processing photographs and for all three types of modeled radiation (Figure 3). Traditional models (ELLIPSE and POLYGON.GAP) provide error values that are average in comparison to other models. The highest RMSE value was usually observed for POLYGON.LiDAR. Taking into account in modeling the diverse height of trees surrounding the gap (LiDAR.0) lowered the value of RMSE error. For subsequent models (LiDAR.1 to LiDAR.5), a gradual decrease of RMSE value was observed. Only the LiDAR.3, LiDAR.4, and LiDAR.5 models were an exception for the UNVAR method, where RMSE value increases.

An analogous dependence of the percent error (PBIAS) on the gap shape model was observed (Figure 4) for various types of radiation and various methods of photograph processing. Also in this comparison the PBIAS values for the LiDAR.3, LiDAR.4, and LiDAR.5 models for the UNVAR method diverge from the general tendency and have positive values.

Apart from that, a clear trend for decreasing amplitude between the highest and lowest error for comparison of GLA and SRT models was observed along with the increasing gap sizes in models based on LiDAR data (Figure 3). The strongest decrease of this difference was observed for LiDAR.0 to LiDAR.3 models of gap shapes.

### 3.3. Relation between the Difference of Outcomes of the GLA and SRT Models and the Distance from Sampling Points to the Nearest Tree Trunk

The relation between the distance to the nearest tree and the value of outcome divergence between the GLA and SRT models for different strategies of solar radiation modeling has a distinct character (Figure 4). For most methods of photograph processing and analyzed types of radiation, a negative correlation was observed ($R$—0.7 to −0.4). By contrast, very low correlation coefficient values were observed for selected gap shape models (LiDAR.3 and LiDAR.4) in the UNVAR method and in the AUTOVAR method (LiDAR.4 and LiDAR.5). This phenomenon is particularly desirable in modeling because such results do not point to spatially correlated errors.

## 4. Discussion

### 4.1. Exposure and Thresholding

Authors of various publications dedicated to the modeling of radiation conditions at the forest floor use results obtained on the basis of hemispherical photograph analysis carried out using GLA or other programs for verification of their models [58–61]. However, the impact that exposure or thresholding [58] used for the processing of photographs can have on assessing the usefulness of a model is seldom a subject of study. Multiple publications indicate that the evaluation of gap fractions in the canopy, and hence also of the radiation conditions at the forest floor, depends to a large degree on the method of photograph exposure [62,63]. Zhang et al. [64] suggest that automatic exposure, in the case of canopies with bigger openness, can result in the disappearance of numerous small openings from the image of the canopy, i.e., in underestimation of canopy openness. Photographs of

the canopy taken in the studied artificial gap are an example of extremely great canopy openness (c.f. Table 1), much larger than in cited studies. In contrast to photographs taken under the closed canopy of a forest stand, in the case of photographs taken in artificial gaps the lateral well-lit fragments of the tree crowns are more often visible against the sky, and less often the shadowed lower parts. Automatic exposure gives the opposite effect in this situation: it does not conceal certain openings but hides smaller fragments of the crowns (Figure 5). Strongly-lit outer fragments of the crowns are classified as the open sky during thresholding.

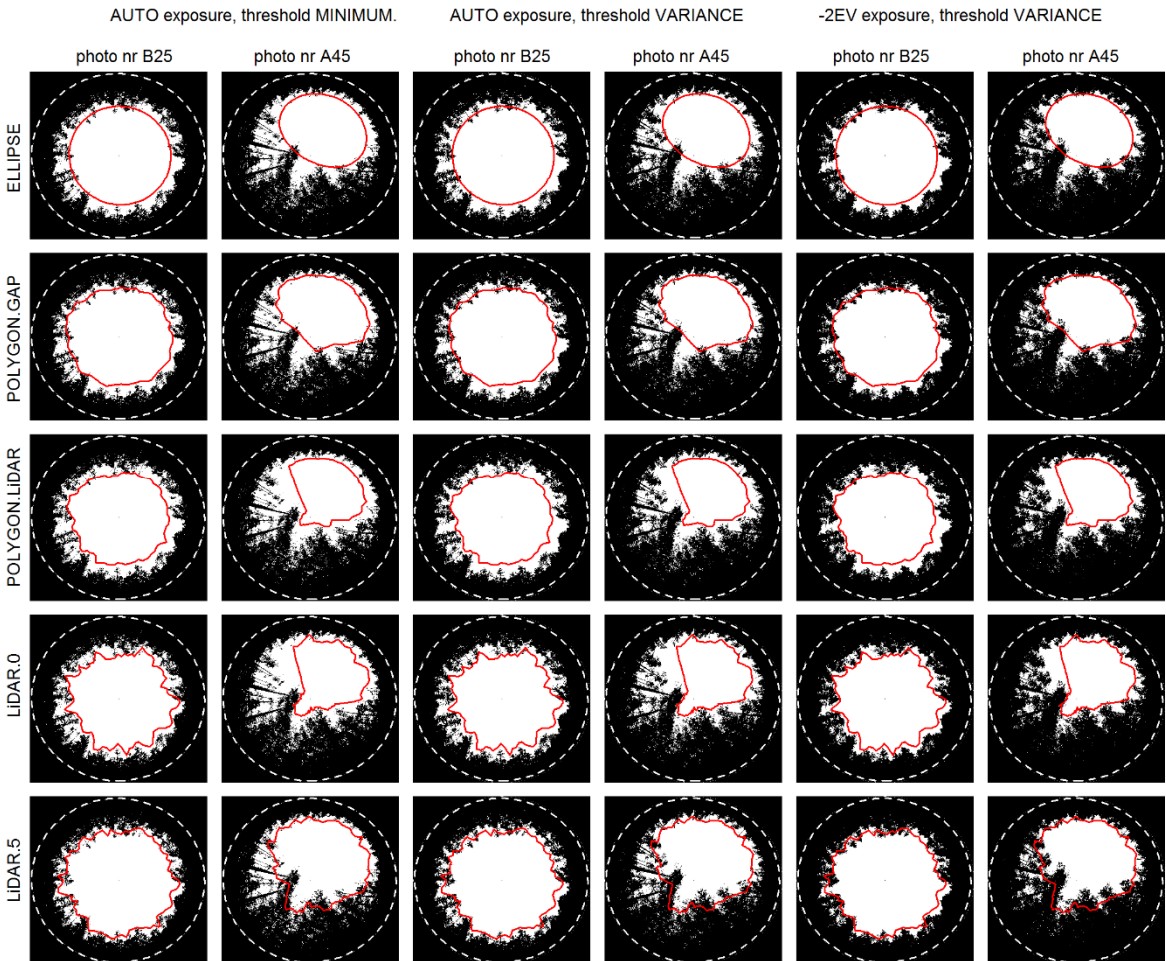

**Figure 5.** Comparison of the top edge of the artificial gap visible on hemispherical photographs processed in different ways (AUTOMIN, AUTOVAR, UNVAR) with the top edge of the gap (ViewShed—marked by the red line) generated for DSMs representing various gap shape models. Two sampling points located in the center (B25) and at the artificial gap's edge (A45) are presented.

Automatic thresholding eliminates subjectivism in decision-making for particular photographs [49]; however, the features of particular thresholding algorithms may significantly impact the results of radiation conditions estimation [47,48]. The change of the MINIMUM algorithm to the VARIANCE algorithm while using automatic exposure resulted in better visibility of the fragments of tree crowns surrounding the gap that entered the picture in black-and-white images of the canopy. This caused a decrease of canopy openness as calculated by the GLA program (Table 1). In the case of the VARIANCE algorithm, the percent bias (PBIAS) decreased in comparison to the GLA and SRT models by on average 4.12 percentage points with small fluctuations related to gap shape model and type of modeled radiation.

The application of underexposure (−2 EV) in the same method of thresholding resulted in a further decrease of openness but also in better visibility of tree crowns surrounding the artificial gap. These results indicate that underexposure of photographs in artificial gaps by a −2 EV value is a better solution than automatic exposure. This result is confirmed by studies of other authors, carried out in forest stands with high canopy openness [62]. The application of underexposure of photographs resulted in a decrease in percent bias (PBIAS) in comparison to GLA and SRT models on average by 8.63 percentage points in reference to the AUTOVAR method and by 12.75 percentage points in reference to the AUTOMIN method. Studies by Glatthorn and Beckschäfer [48] indicate that the VARIANCE algorithm better detects fragments of tree crowns during the processing of photographs that do not have overexposed parts, hence decreased exposure could improve its performance. Therefore, it can be stated that hemispherical photographs taken in the mode of manual exposure should not be applied as a point of reference for models of light conditions in artificial gaps created for silvicultural applications. Underexposure of photographs by −2 EV improves the detection of fragments of crowns of the trees surrounding the gap, which can help to obtain a more precise estimation of radiation conditions in gaps.

### 4.2. Modeled Gap Shape

The impact of the method of gap shape modeling on the assessment of congruity of GLA and SRT models is best discussed on the basis of results from the modeling of diffuse radiation. Diffuse light reaches the gap floor from all sectors of the sky and not only from its southern part (the situation in earth's northern hemisphere). For this reason, the shape of the top gap edge has an impact on the transmissivity of diffuse radiation into the gap along the whole circumference. It can be expected that the higher the congruence of the top edge of the gap as analyzed by DSM (ViewShed) with the top edge of the gap in the hemispherical photograph, the higher the congruence between GLA and SRT models.

Traditional gap shape models (ELLIPSE, POLYGON.GAP) used in the modeling of radiation conditions in gaps were assessed as better (taking into account PBIAS values, RMSE and $R^2$) than the model in which the shape of the gap base was estimated through the direct interpretation of the results of laser scanning (LiDAR.GAP). It is, however, important to emphasize that the choice of size and location of the ellipse representing the base of gap shape was carried out using a detailed map of tree trunks surrounding the gap. Without the time-consuming gathering of such data, matching of the ELLIPSE model would probably be worse, and construction of the POLYGON.GAP model would not be possible at all.

A boundary of the gap base that is approximated to a vertical projection of the crown edges (Figure 1b), which is a basis for the LiDAR.GAP model was the worst of all nine analyzed models, and a high negative PBIAS value indicated a significant underestimation of radiation conditions. The LiDAR.0 model, which takes into account the height differences of the trees surrounding the gap, displayed decidedly smaller errors than LiDAR.GAP, but it still estimated radiation conditions as lower than the traditional models (ELLIPSE, POLYGON.GAP) did. Expanding the gap base, in connection with the application of information about the height differences of the trees surrounding the gap (LiDAR.1 to LiDAR.5 models), caused the SRT model to provide results closer to those of GLA. In the case of photographs with automatic exposure, it resulted in a reduction of PBIAS nearly to zero. However, in photographs taken with manual correction of exposure, LiDAR.4 and LiDAR.5 models provided even positive PBIAS values. None of the nine 3D gap models made it possible to obtain the shape of the top gap edge (ViewShed) perfectly congruent with the gap edge visible on the hemispherical photograph (Figure 5).

In subsequent LiDAR models, ViewShed (the top edge of the modeled gap) approaches the real gap edge visible in the photograph. In the case of the LiDAR.5 model, ViewShed even transects the image of the tree crowns surrounding the gap. In the case of photographs with exposure correction, the smallest absolute PBIAS value was obtained for the extension by 3 m (LiDAR.2 model). In the case of extensions of 2 or more meters (LiDAR.2, LiDAR.3, and LiDAR.4 models) a clear decrease in

amplitude between the maximum and minimum error value was observed (c.f. Figure 3), even below
the fluctuations observed in traditional models (ELLIPSE, POLYGON.GAP). The advantage of gap
model LiDAR.3 above other LiDAR based models could be explained in the following way. Let us
assume that an observer stays in the close proximity of trees growing on the edge of a gap. When the
observer is looking to the zenith, the crown of the neighboring tree obscures the substantial part of the
sky and the border of the non-obscured sky is former by the twigs growing on the side of the tree not
of its top. The sky limit is formed by the vertical projection of the tree crown. But the same observer
looking at the opposite part of a gap will see that border between obscured and the non-obscured sky
is lying rather on twigs growing from treetops than from their sides (it could be seen on hemispherical
photographs taken on point A45 (Figure 5). The sky limit is formed by the side view (silhouette) of
the tree crown. The more distant is an observer from a tree the more important in obscuring the sky
are twigs growing from the tops of trees. The majority of points where hemispherical photographs
were taken were placed far from the gap border and this is why is so important to use LiDAR points
lying on the parts of the crown forming tree silhouette to delimitate sky limit. From a geometrical
point of view silhouette in its shape resemble vertical crosscut of tree crown running by vertical tree
axis. During the creation of subsequent models LiDAR.0, LiDAR.1, LiDAR.2, LiDAR.3, LiDAR.4,
LiDAR.5 we virtually cut three dimensional shapes of tree crown surrounding gaps described by
LiDAR. Our results suggest that crosscuts made by LiDAR.2 and LiDAR.3 models give the best results
(they resemble the trees silhouettes in the best way) and we think it is because this cut runs in the
closest proximity of vertical axis of trees (Figure 6). That is why DSM model based on a vertical
projection of crown trees extended in every direction about the distance comparable with mean crown
radius could work also in other localizations.

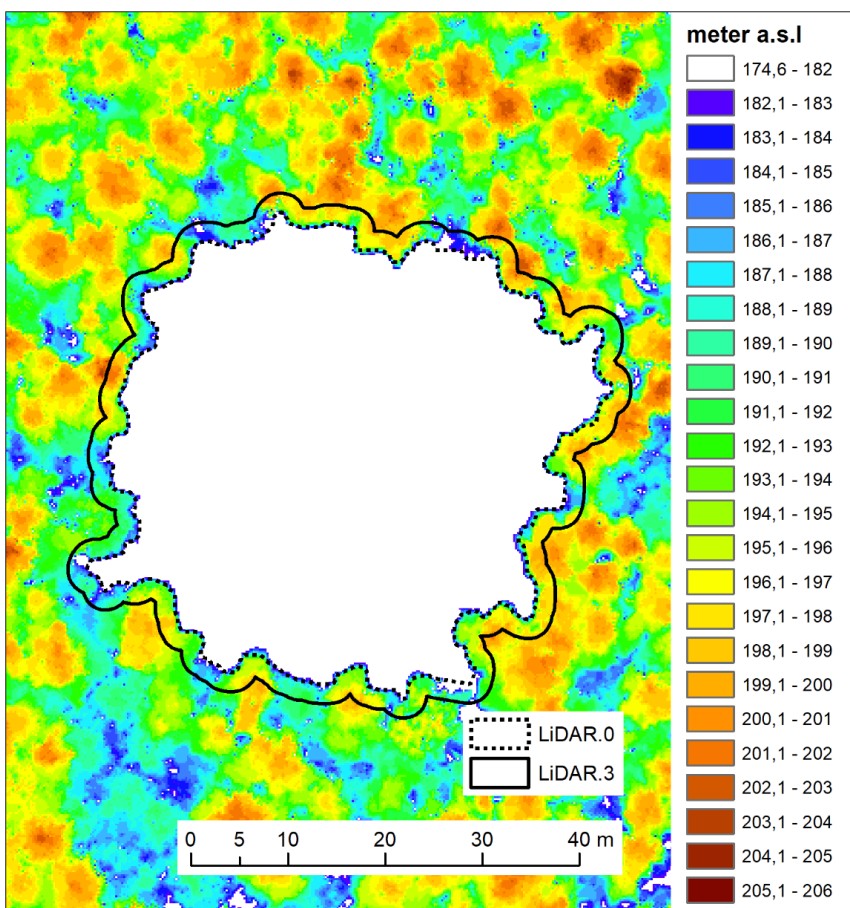

**Figure 6.** Visualization of forest canopy around investigated gap made on the basis of LiDAR data.
The dashed line depicts the gap border in the LiDAR.0 model and solid line in the LiDAR.3 model.

## 5. Conclusions

In the case of the studied gap, a mean horizontal distance between the tree trunk and the echo from laser scanning registered at the circumference of its crown was about 3 m. The SRT model build on DSM in which the base of the gap was extended on 3 m provides results that are the closest to the data obtained from the GLA program using photographs after optimal processing. Basing on the congruence of GLA and SRT models it can be stated that the optimum value for extension of the gap base for the LiDAR dependent models corresponds with the dimensions of the tree crowns surrounding the gap.

One of the big advantages of LiDAR data is the possibility of a detailed description of 3D forest canopy structure which is very useful in the practical implementation of modeling. This is also a very fast and accurate technology comparing to classical surveying methods. It is especially valuable in a forest environment where accurate filed measurements are strongly limited. There is no other technique that allows the use of such accurate models for vast areas. An additional advantage of the described solution is an easy implementation in popular GIS software.

Taking into account increasing accessibility of LiDAR data lowering costs and growing operational simplicity could be expected. In our study, we used publicly available LiDAR data collected for other purposes. This solution substantially reduces costs however in the case when such LiDAR scanning is not frequently done it could be seen as factor constraining implementation of our method. This problem can be easily solved by unmanned aerial vehicles that are suitable for spatial data capture of small areas such as analyzed in this research. Strong congruence of canopy height models build on data taken from unmanned aerial vehicles and aircraft is observed [31,65] so there is no doubt that this method can be successfully used for modeling gap shape.

Our method could be recommended for situations when the border between the gap and non-gap area is well defined like in dense monolayer forest stands. It can be easily transferable not only to artificial gaps but for any empty areas in the forest as well. It is specially important in cases where fast and positive trees growing is expected. Bigger distances between trees in sparse stands or the presence of high diversity of trees heights may potentially affect the results of our modeling approach because may increase the chance that solar radiation will reach forest floor in the gap through the surrounding canopy. Availability of the open access data from laser scanning offered in many countries [66] and the possibility of alternative acquisition of DSMs from unmanned platforms [67] can promote the implementation of the proposed solution for modeling light conditions in artificial gaps.

**Author Contributions:** Conceptualization, L.B. and M.B.; methodology, L.B. and M.B.; software, L.B. and M.B.; validation, L.B. and M.B..; formal analysis, L.B.; investigation, L.B.; resources, L.B. and M.B.; writing—original draft preparation, L.B. and M.B.; writing—review and editing, L.B. and M.B.; visualization, L.B. and M.B.; supervision, L.B. and M.B. All authors have read and agreed to the published version of the manuscript.

**Funding:** This research received no external funding.

**Conflicts of Interest:** The authors declare no conflict of interest.

## Appendix A

**Table A1.** Numerical values used to create Figure 3.

| Radiation | Threshold Exposition | Measure of Discrepancy between Models | Variant of SOLAR Model | | | | | | | | |
|---|---|---|---|---|---|---|---|---|---|---|---|
| | | | ELLIPSE | POLYGON.GAP | POLYGON.LiDAR | LiDAR.0 | LiDAR.1 | LiDAR.2 | LiDAR.3 | LiDAR.4 | LiDAR.5 |
| **Total** | AUTOMIN | minimal absolute error | −276.64 | −270.26 | −497.96 | −333.18 | −305.29 | −229.95 | −191.65 | −151.94 | −117.55 |
| | | mean absolute error | −139.19 | −113.91 | −210.35 | −157.22 | −139.81 | −106.55 | −72.83 | −39.12 | −8.38 |
| | | maximal absolute error | −49.72 | −25.27 | -95.79 | −74.12 | −58.4 | −23.91 | 13.73 | 50.77 | 79.73 |
| | | RMSE | 154.88 | 136.37 | 229.24 | 173.3 | 154.88 | 118.67 | 86.96 | 58.93 | 41.94 |
| | AUTOVAR | minimal absolute error | −238.23 | −233.62 | −424.99 | −260.21 | −232.32 | −143.93 | −88.43 | −49.19 | −27.86 |
| | | mean absolute error | −105.57 | −80.29 | −176.73 | −123.6 | −106.19 | −72.94 | −39.22 | −5.5 | 25.24 |
| | | maximal absolute error | −23.93 | −1.19 | −70 | −48.32 | −42.77 | −22.15 | 3.9 | 28.46 | 62.82 |
| | | RMSE | 121.71 | 105.57 | 194.44 | 136.13 | 117 | 79.74 | 47.32 | 23.01 | 34.43 |
| | UNVAR | minimal absolute error | −162.86 | −155.25 | −367.91 | −203.13 | −175.23 | −62.42 | −14.46 | 13.31 | 33.99 |
| | | mean absolute error | −45.25 | −19.98 | −116.42 | −63.28 | −45.87 | −12.62 | 21.1 | 54.82 | 85.56 |
| | | maximal absolute error | 19.12 | 41.86 | −26.95 | −0.77 | 12.67 | 32.52 | 64.72 | 95.56 | 142.55 |
| | | RMSE | 68.94 | 63.18 | 138.48 | 80.53 | 62.6 | 28.66 | 31.43 | 59.54 | 90.56 |
| **Direct** | AUTOMIN | minimal absolute error | −179.32 | −186.03 | −241.06 | −169.07 | −162.97 | −154.95 | −122.26 | −76.76 | −43.38 |
| | | mean absolute error | −64.92 | −54.83 | −94.83 | −67.24 | −58.88 | −43.56 | −27.46 | −11.37 | 5.28 |
| | | maximal absolute error | −6.18 | 3.78 | −16.66 | −3.47 | −0.76 | 2.5 | 15.33 | 34.26 | 61.27 |
| | | RMSE | 82.12 | 78.77 | 111.42 | 78.37 | 69.5 | 55.1 | 40.59 | 29.05 | 24.04 |
| | AUTOVAR | minimal absolute error | −149.07 | −158.88 | −205.22 | −127.83 | −108.6 | −100.58 | −67.89 | −27.31 | −11.34 |
| | | mean absolute error | −50.48 | −40.39 | −80.39 | −52.8 | −44.44 | −29.12 | −13.02 | 3.06 | 19.72 |
| | | maximal absolute error | 4.16 | 14.12 | -6.32 | 6.82 | 9.53 | 12.78 | 24.43 | 32.98 | 57.16 |
| | | RMSE | 68.76 | 66.62 | 98.37 | 63.8 | 54.11 | 38.8 | 24.83 | 17.89 | 26.33 |
| | UNVAR | minimal absolute error | −108.99 | −119.92 | −159.63 | −81.64 | −62.45 | −51.44 | −18.75 | 0.16 | 10.1 |
| | | mean absolute error | −21.1 | −11.01 | −51.02 | −23.42 | −15.06 | 0.26 | 16.36 | 32.44 | 49.09 |
| | | maximal absolute error | 22.11 | 32.07 | 11.63 | 29.44 | 32.15 | 36.95 | 57.65 | 76.7 | 103.35 |
| | | RMSE | 43.71 | 46.04 | 71.37 | 37.55 | 28.84 | 20.46 | 25.1 | 37.76 | 54.51 |
| **Diffuse** | AUTOMIN | minimal absolute error | −148.33 | −127.2 | −298.95 | −213.83 | −216.96 | −153.22 | −26.94 | −97.13 | −74.17 |
| | | mean absolute error | −74.27 | −59.09 | −115.52 | −89.98 | −80.93 | −63 | −45.38 | −27.75 | −13.66 |
| | | maximal absolute error | −30.45 | −14.8 | -59.16 | −39.89 | −33.66 | −19.48 | −1.61 | 16.51 | 30.51 |
| | | RMSE | 79.13 | 65.22 | 125.9 | 100.13 | 90.79 | 69.08 | 51.85 | 35.42 | 24.31 |
| | AUTOVAR | minimal absolute error | −93.39 | −90.26 | −246.3 | −161.18 | −164.31 | −67.78 | −49.11 | −31.98 | −16.52 |
| | | mean absolute error | −55.09 | −39.9 | −96.34 | −70.8 | −61.75 | −43.82 | −26.2 | −8.57 | 5.53 |
| | | maximal absolute error | −28.08 | −15.31 | −58.12 | −31.37 | −27.52 | −18.55 | −6.68 | 9.66 | 27.35 |
| | | RMSE | 57.55 | 43.86 | 103.02 | 76.5 | 67.15 | 45.34 | 27.76 | 11.97 | 11.42 |
| | UNVAR | minimal absolute error | −55.77 | −52.65 | −208.28 | −123.16 | −126.28 | −29.1 | −11.31 | 7.27 | 17.75 |
| | | mean absolute error | −24.15 | −8.96 | −65.4 | −39.86 | −30.81 | −12.88 | 4.74 | 22.37 | 36.47 |
| | | maximal absolute error | −2.57 | 17.01 | −33.73 | −7.96 | −4.11 | 4.85 | 24.15 | 47.15 | 64.84 |
| | | RMSE | 28.54 | 19.37 | 73.43 | 46.89 | 38.38 | 15.74 | 9.86 | 24.66 | 38.85 |

Exposure & Thresholding coding: AUTOMIN—auto exposure (−2EV) & MININMUM thresholding algorithm; AUTOVAR—auto exposure & VARIANCE thresholding algorithm; UNVAR—underexposure (−2EV) & VARIANCE thresholding algorithm.



**Table A2.** Numerical values used to create Figure 4.

| Radiation | Threshold Exposition | Measure of Dependence between Models | Variant of SOLAR Model | | | | | | | | |
|---|---|---|---|---|---|---|---|---|---|---|---|
| | | | ELIPSE | POLYGON GAP | POLYGON LiDAR | LiDAR.0 | LiDAR.1 | LiDAR.2 | LiDAR.3 | LiDAR.4 | LiDAR.5 |
| Total | AUTOMIN | R2 | 0.90 | 0.91 | 0.76 | 0.84 | 0.84 | 0.87 | 0.87 | 0.86 | 0.85 |
| | | PBIAS (%) | −19.10 | −15.70 | −28.90 | −21.60 | −19.20 | −14.60 | −10.00 | −5.40 | −1.20 |
| | | r (edge) | −0.68 | −0.65 | −0.68 | −0.81 | −0.78 | −0.70 | −0.63 | −0.59 | −0.47 |
| | AUTOVAR | R2 | 0.93 | 0.93 | 0.82 | 0.93 | 0.94 | 0.97 | 0.97 | 0.97 | 0.95 |
| | | PBIAS (%) | −15.20 | −11.60 | −25.50 | −17.80 | −15.30 | −10.50 | −5.70 | −0.80 | 3.60 |
| | | r (edge) | −0.49 | −0.47 | −0.56 | −0.74 | −0.72 | −0.61 | −0.47 | −0.38 | −0.26 |
| | UNVAR | R2 | 0.94 | 0.94 | 0.83 | 0.94 | 0.94 | 0.97 | 0.97 | 0.96 | 0.94 |
| | | PBIAS (%) | −7.10 | −3.20 | −18.40 | −10.00 | −7.20 | −2.00 | 3.30 | 8.70 | 13.50 |
| | | error/distance cor. | −0.41 | −0.45 | −0.49 | −0.68 | −0.65 | −0.52 | 0.10 | −0.12 | −0.44 |
| Direct | AUTOMIN | R2 | 0.93 | 0.93 | 0.85 | 0.93 | 0.95 | 0.95 | 0.94 | 0.93 | 0.93 |
| | | PBIAS (%) | −21.20 | −17.90 | −31.00 | −22.00 | −19.20 | −14.20 | −9.00 | −3.70 | 1.70 |
| | | error/distance cor. | −0.51 | −0.51 | −0.45 | −0.66 | −0.66 | −0.58 | −0.58 | −0.63 | −0.27 |
| | AUTOVAR | R2 | 0.94 | 0.93 | 0.85 | 0.95 | 0.97 | 0.98 | 0.98 | 0.97 | 0.96 |
| | | PBIAS (%) | −17.30 | −13.80 | −27.60 | −18.10 | −15.20 | −10.00 | −4.50 | 1.00 | 6.80 |
| | | error/distance cor. | −0.41 | −0.42 | −0.35 | −0.58 | −0.61 | −0.54 | −0.51 | −0.30 | −0.03 |
| | UNVAR | R2 | 0.96 | 0.96 | 0.88 | 0.96 | 0.98 | 0.98 | 0.97 | 0.96 | 0.94 |
| | | PBIAS (%) | −8.00 | −4.20 | −19.40 | −8.90 | −5.70 | 0.10 | 6.20 | 12.40 | 18.70 |
| | | error/distance cor. | −0.40 | −0.40 | −0.33 | −0.62 | −0.60 | −0.33 | 0.12 | 0.16 | −0.14 |
| Diffuse | AUTOMIN | R2 | 0.80 | 0.78 | 0.64 | 0.71 | 0.66 | 0.70 | 0.69 | 0.68 | 0.67 |
| | | PBIAS (%) | −17.60 | −14.00 | −27.40 | −21.40 | −19.20 | −15.00 | −10.80 | −6.60 | −3.20 |
| | | error/distance cor. | −0.75 | −0.73 | −0.71 | −0.73 | −0.67 | −0.60 | −0.52 | −0.45 | −0.33 |
| | AUTOVAR | R2 | 0.91 | 0.89 | 0.86 | 0.93 | 0.90 | 0.96 | 0.96 | 0.96 | 0.95 |
| | | PBIAS (%) | −13.70 | −9.90 | −24.00 | −17.60 | −15.40 | −10.90 | −6.50 | −2.10 | 1.40 |
| | | error/distance cor. | −0.62 | −0.54 | −0.69 | −0.75 | −0.65 | −0.59 | −0.31 | 0.05 | −0.58 |
| | UNVAR | R2 | 0.91 | 0.88 | 0.85 | 0.93 | 0.90 | 0.97 | 0.96 | 0.96 | 0.96 |
| | | PBIAS (%) | −6.50 | −2.40 | −17.60 | −10.70 | −8.30 | −3.50 | 1.30 | 6.00 | 9.80 |
| | | error/distance cor. | −0.36 | −0.44 | −0.61 | −0.69 | −0.54 | −0.26 | −0.40 | −0.57 | −0.72 |

Exposure & Thresholding coding: AUTOMIN—auto exposure (−2EV) & MININMUM thresholding algorithm; AUTOVAR—auto exposure & VARIANCE thresholding algorithm; UNVAR—underexposure (−2EV) & VARIANCE thresholding algorithm.

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
