# Peer review of "Application of LiDAR Data for the Modeling of Solar Radiation in Forest Artificial Gaps—A Case Study"

_forests, doi:10.3390/f11080821_

Round 1

Reviewer 1 Report

The paper subject is original and new, moreover has a potential of development theoretical and applied.

The forest regeneration is an actual topic due to climate change and clear cutting. The use of  LIDAR data for forest gaps modeling and results assessment is a challenge.

The use of automatic threshold for camera exposure like AUTOVAR, AUTOMIN, and UNVAR eliminate the subjectivism and make the methodology more replicable.

The model gap obtained by LIDAR data are more sharpen that the reference one given by camera, but the potential is really huge if new machine learning algorithms will be adapted for this propose. 

The abstract is concise and relevant, the methodology is clear structured, the results are well introduced and the discussion are in a proper form.

Author Response

We are grateful to the Reviewer for so positive comments and conclusions. We are really happy that the reviewer accepted the whole text in the present form and do not address any corrections. We declare that after minor revision we improved the text taking into account grammar and some small mistakes in the text. We hope that in this form it will fill the expatiations of potential readers. We also declare that text will be checked by the MDPI English editing department after acceptance.

Reviewer 2 Report

The study presents an interesting experiment and the data worth publishing, however, the submitted manuscript (MS) requires several changes before it can be published in a scientific journal.

L2: the MS is unambiguously a „case study“ – this is not a problem per se, anyway it needs to be indicated already in the title

L22: GLA is abbreviation of „Gap light analyzer“. Please, insert this full name and add the reference: „Frazer, G.W., Canham, C.D., and Lertzman, K.P. 1999. Gap Light Analyzer (GLA), Version 2.0: Imaging software to extract canopy structure and gap light transmission indices from true-colour fisheye photographs, user’s manual and program documentation. Copyright © 1999: Simon Fraser University, Burnaby, British Columbia, and the Institute of Ecosystem Studies, Millbrook, New York. “

L98–100: you wrote, that the main goal of the study was „to propose simpler method...for estimation of radiation conditions in gaps“. In fact, the study deals with the comparison of the indirect estimation of radiation using the hemispherical photographs (GLA) and the estimation of radiation using the model implemented in ArcGIS (Solar radiation tool – SRT) that uses variable delineation of artificially created gap. The goal should be reworded to correspond with the study content.

L107: you present the value of mean crown radius to be „around 3 meter“, on the other hand you give the exact value of SD (1.36 m). This is quite awkward, you should present the exact value of mean as well.

L115: in Figure 1B, the line delineating one of analyzed boundaries of the base of 3-D shape representing the gap (either POLY.LiDAR or LiDAR 0) is missing

L143: you state that the laser scanning data originate from 2013. When the hemispherical photographs were taken? In the case that the time between scanning and fish-eye imaging was relatively long, what was the lateral increment of the crowns of the edge trees? Wasn´t there a significant impact of the open gap reduction on the decrease of the radiation values obtained from hemispherical photographs?

L225–226: What kind of correction you performed „regard to the real projection of the Sigma 8 mm objective“? Was it the correction of 8 mm lens on 180 deg. view angle?

L256: add the meaning of abbreviation RMSE (root mean square error)

L258: in all formulas, use „SRT“ instead of „Solar“

L270–273: the entire paragraph belongs to the chapter „Methods“

L275–276: present the results of the t-test (e.g. in Tab.1 in form of different letters)

L 403: it is strange that in the entire subchapter 4.2 there is no single reference to some relevant scientific source. Is there really no published study that could support e.g. the idea presented in L440 and following lines?

L504: in column Threshold /Exposition use the names AUTOMIN, AUTOVAR and UNVAR

Author Response

Response to Reviewer 2 Comments

We are grateful to the Reviewer for valuable comments. The text has been proofread again and we have incorporated most of the suggestions. We also declare that text will be checked by MDPI English editing department after acceptance.

Point 1: L2: the MS is unambiguously a „case study“ – this is not a problem per se, anyway it needs to be indicated already in the title

Response 1: We had agreed with the reviewer and extended the manuscript title in the following way: “Application of LiDAR data for the modeling of solar radiation in forest artificial gaps - a case study”

Point 2: L22: GLA is abbreviation of „Gap light analyzer“. Please, insert this full name and add the reference: „Frazer, G.W., Canham, C.D., and Lertzman, K.P. 1999. Gap Light Analyzer (GLA), Version 2.0: Imaging software to extract canopy structure and gap light transmission indices from true-colour fisheye photographs, user’s manual and program documentation. Copyright © 1999: Simon Fraser University, Burnaby, British Columbia, and the Institute of Ecosystem Studies, Millbrook, New York. “

Response 2: We added the full name of GLA software in the indicated fragment of Abstract. The mentioned software is already cited in our improved version of the manuscript in line 229 and in References in 54 positions than addition of this citation is not needed.

Point 3: L98–100: you wrote, that the main goal of the study was „to propose simpler method...for estimation of radiation conditions in gaps“. In fact, the study deals with the comparison of the indirect estimation of radiation using the hemispherical photographs (GLA) and the estimation of radiation using the model implemented in ArcGIS (Solar radiation tool – SRT) that uses variable delineation of artificially created gap. The goal should be reworded to correspond with the study content.

Response 3: We reformulated the description of our main study goal in line with the Reviewer suggestion. Now this part of manuscript is: “The main goal of this study is was comparison of the indirect estimation of radiation using the hemispherical photographs (GLA) and the estimation of radiation using the model implemented in ArcGIS (Solar radiation tool – SRT) that uses variable delineation of the artificially created gap. On this basis we proposed a simple and reliable procedure enabling estimation of radiation conditions in artificial gaps which can be easily implemented by ready to use model in popular GIS software.”

Point 4: L107: you present the value of mean crown radius to be „around 3 meter“, on the other hand you give the exact value of SD (1.36 m). This is quite awkward, you should present the exact value of mean as well.

Response 4: We agree with the Reviewer that simultaneously referring to the approximate value of a variable and providing its standard deviation is quite awkward. In the improved version of the manuscript we reported a mean value of the mentioned variable.

Point 5: L115: in Figure 1B, the line delineating one of analyzed boundaries of the base of 3-D shape representing the gap (either POLY.LiDAR or LiDAR 0) is missing

Response 5: The 3-D shape shapes representing POLY.LiDAR and LiDAR 0 have exactly the same basis as it was written in the previous version of the manuscript in the lines 194 -200. During the preparation of figure 1B  the symbol of POLY.LiDAR was “covered” by the symbol of LiDAR 0. We improved the figure to correct this mistake. The basis of POLY.LiDAR shape is depicted with a yellow solid line and the LiDAR 0 with a dashed blue line as could be seen below:

Point 6: L143: you state that the laser scanning data originate from 2013. When the hemispherical photographs were taken? In the case that the time between scanning and fish-eye imaging was relatively long, what was the lateral increment of the crowns of the edge trees? Wasn´t there a significant impact of the open gap reduction on the decrease of the radiation values obtained from hemispherical photographs?

Response 6: Hemispherical photographs were taken on May 28, 2013. This information was added to the improved version of the manuscript. Older Scots pine trees (investigated forest stand was around 102 years old) are known from the very limited tendency to expand their crowns. On the other hand, broadleaved trees could expand their crowns in the direction of artificial gap centres more vigorously especially when they are young. Fortunately, on the gap border grow only a few old birches (Betula pendula) and we think that their crown expansion also was not very big and could be negligible for the results of the analysis.

Point 7: L225–226: What kind of correction you performed „regard to the real projection of the Sigma 8 mm objective“? Was it the correction of 8 mm lens on 180 deg. view angle?

Response 7: During the analysis hemispherical photographs are interpreted by GLA algorithm as a map of the hemisphere above the point where the photograph was taken. The polar coordinates of canopy elements (e.g. a branch) on the digital images are transformed to 3D coordinates of the mentioned element to assess if it could obscure the solar disc on a particular date and time of the year. It is possible only when the projection of the used lens is known. The projection could be interpreted as a mathematical formula that describes how the coordinates of an object on a hemisphere are transformed into coordinates of an object on the plane. The manufactures usually provide very general information about kind of projection is implemented in their lenses. Usually, due to some issues connected with lens production real projection is not exactly the same as declared by manufacturer theoretical projection. This problem was discovered long ago and the manual of GLA software (referenced in position 54) describes how to correct discrepancy between the real and theoretical projection of used lenses. The description of discrepancy observed in our lens (Sigma 8 mm F3.5 EX DG ) was furnished by the authors cited in reference position 55. The implemented correction has practical meaning for an object on a hemisphere with smaller zenith angles suggesting that they are on certain height above the horizon and could cast a shadow on the place where photograph was taken. Objects visible on the periphery hemispherical image have a zenith angle close to 90 degrees and their influence on light conditions at the investigated point is very limited.

Point 8. L256: add the meaning of abbreviation RMSE (root mean square error)

Response 8: We added the meaning of abbreviation RMSE in the improved version of manuscript.

Point 9. L258: in all formulas, use „SRT“ instead of „Solar“

Response 9: We changed the mentioned formulas according to Reviewer suggestion.

Point 10. L270–273: the entire paragraph belongs to the chapter „Methods“

Response 10: The content of Table 1 present the results. We can agree with Reviewer that sentence in lines 272-273 “Canopy openness was chosen as a synthetic indicator of the influence of different hemispherical photography exposure and thresholding scenarios on assessment of forest canopy properties” sound like a part of methods chapter. However, in our opinion, this sentence is more needed at the beginning of this chapter because it is more useful for the potential readers here than if it had been inserted elsewhere in the methods section.

Point 11. L275–276: present the results of the t-test (e.g. in Tab.1 in form of different letters)

Response 11: We added in Table 1 letters indicating significantly different means according to Reviewer suggestion as well we add a short explanation of the meaning of the mentioned letters in the improved manuscript.

Point 12. L 403: it is strange that in the entire subchapter 4.2 there is no single reference to some relevant scientific source. Is there really no published study that could support e.g. the idea presented in L440 and following lines?

Response 12: To our best knowledge, our solution explained in detail in chapter 4.2 was never published before. The novelty of this solution is the main reason why we do not refer to other papers. We just try to explain in detail how our solution works. There are quite numerous articles describing the use of 3D model to describe the shape of the artificial gap and briefly they were mentioned in the introductory part of our manuscript. Those models as a base of an artificial gap utilize basic geometrical shapes like an ellipse or polygon in which vertices are placed on the bases of surrounding trees. The idea of the utilization of information of LiDAR data to model light conditions in gaps is not new. Published models of this kind utilize LiDAR data to trace sun rays penetrating crowns of trees. It is very computing demanding technology and of course could have a clear advantage in the case of light modeling in under canopy conditions. In the case of artificial gaps created for silvicultural purposes the amount of light passing over the crowns of the surrounding trees is very small in comparison with the light passing over the huge gap in forest canopy created by artificial gap creation. In such a specific situation, the proper description of the upper border of such canopy gap is much more important that estimation of light coming from lower zenith angles.

Point 13. L504: in column Threshold /Exposition use the names AUTOMIN, AUTOVAR and UNVAR

Response 11: We changed in Appendix table acronyms according to Reviewer suggestion.
